# The Effect of the Optimization Selection of Position Analysis Route on the Forward Position Solutions of Parallel Mechanisms

**Huiping Shen \*, Qing Xu, Ju Li and Ting-li Yang**

School of Mechanical Engineering, Changzhou University, Changzhou 213164, China; xq950220@163.com (Q.X.); wangju0209@163.com (J.L.); yangtl@126.com (T.-l.Y.)

**\*** Correspondence: shp65@126.com

**Abstract:** The forward position solution (FPS) of any complex parallel mechanism (PM) can be solved through solving in sequence all of the independent loops contained in the PM. Therefore, when solving the positions of a PM, all independent loops, especially the first loop, must be correctly selected. The optimization selection criterion of the position analysis route (PAR) proposed for the FPS is presented in this paper, which can not only make kinematics modeling and solving efficient but also make it easy to get its symbolic position solutions. Two three-translation PMs are used as the examples to illustrate the optimization selection of their PARs and obtain their symbolic position solutions.

**Keywords:** parallel mechanism; forward kinematics; coupling degree

## 1. Introduction

The forward position solution (FPS) of a PM is one of the most important and basic problems in the parallel mechanisms (PMs) research community.

At present, most researchers use the loop vector method [1–3] to establish the input–output position equations of parallel mechanism (PM) and then use numerical or algebraic methods to find its solutions. The numerical method is used directly to solve the position equations, whose advantage is that the real number solutions can be obtained, and the disadvantage is that the iteration is easy to diverge and large calculations are needed. The algebraic methods [4–7], by eliminating the unknowns in the position equations, finally express it as a one-variable higher-order equation, from which all possible solutions can be found. However, algebraic methods need advanced mathematical elimination methods.

It is noticed that the establishment of the above-mentioned position equations of a PM based on the loop vector method does not consider the effect of topological characteristics [8,9] of the PM on its kinematics. For the loop vector method, on the one hand, the loop that some researchers use is a mostly "nature loop" but not actual independent loop. On the other hand, all loops are treated as having equal "ranking status". Therefore, there exist more variables contained in the position equations. Especially when the algebraic method is used, the mathematical elimination processes take more time and are complicated.

For PMs with the same topology of branch chains, there is no need for selecting the optimal position analysis route (PAR). However for PMs with branch chains of different topology, the correct selection of the PAR is necessary, which will affect the efficiency of the FPS and even the solution forms of the forward position (i.e., numerical, closed-form, symbolic). If the PAR is not selected properly, the process of FPS of PMs may be more complicated and difficult, and even their symbolic solutions cannot be obtained. On the contrary, the process of the FPS is convenient, and their symbolic position solutions can be obtained as much as possible.

It can be seen from Ref. [9,10] that the FPS of a PM can be performed by handling its sub-kinematic chains included in the PM in an orderly manner, while an SKC can be solved in accordance with the order of the contained independent loops. Then, for PMs with branch chains of different topologies, the problem of how to sequentially select the first, the $2^{nd}, \dots , v^{\text{-th}}$ independent loops, especially the first loop, will become the key to solving FPS. This problem will directly affect whether the kinematics modeling and FPS can be carried out smoothly.

Through previous work [10,11], the authors find that (1) based on the "principle of least constraint degree" [8,9], there may be multiple topological structure decomposition schemes, which are not all the best PAR. (2) If the PAR is not selected properly, it may make the FPS complicated or impossible. Otherwise, the FPS is simple and convenient, and the symbolic position solutions can be obtained as much as possible. These observations lead the investigation of the paper.

The remainder of the paper is organized as follows. In Section 2, optimization criteria and procedures of optimization selection of PAR are presented. In Section 3, a group of basic formulas of the topological characteristic index used for topological property analysis [12] is described. Two three-translation (3T) PMs with branch chains of different topologies are used as examples to illustrate the optimization selection of PAR and their effect on the efficiency of the FPS and on the symbolic position solutions in Section 4, respectively. Finally, conclusions are drawn in the last part.

## 2. The Optimization Selection Criteria for the Position Analysis Route (PAR)

### 2.1. Optimization Criteria for PAR

In the FPS of the PMs, the principle of the minimum constraint degree [8,9] $\Delta_{\min}$ and the minimum number of independent displacement equations (NIDE) $\xi_{\min}$ should be satisfied at the same time when selecting an optimal PAR. Once choosing the optimal PAR correctly, the FPS can be carried out efficiently, and its symbolic position solutions can be obtained as much as possible.

### 2.2. Procedures of Optimization Selection of PAR

i.   For all loops inside the sub-kinematic chains (SKC) [9] of a PM, the loop with the smallest constraint degree value ($\Delta \geq 0$) should be used as the first loop for the FPS, which can minimize the number of virtual variables [10] that should be assigned when performing FPS to the smallest degree.

ii.  If there are several optional first loops with the same minimum constraint degree $\Delta$, the loop with the smallest of the NIDE $\xi_{\min}$ should be selected as the first loop. In this way, the number of position equations required to solve the loop positions can be minimized, which is exactly equal to $\xi_{\min}$.

iii. If there are both planar $\text{SKC}_{(s)}$ and space $\text{SKC}_{(s)}$ in a PM, the FPS should be started from the planar $\text{SKC}_{(s)}$ first, and then the space $\text{SKC}_{(s)}$ should be analyzed. This is because the NIDE of the planar mechanism loop is always the smallest, i.e., $\xi = 3$.

The above-mentioned criteria and procedures of optimization selection of PAR will be applied to explain the optimization selection of PAR and their effect on the efficiency of the FPS and on the symbolic position solutions of the two three-translation (3T) PMs with branch chains of different topology.

## 3. Basic Formulas for Topological Characteristics Analysis

### 3.1. Analysis of the POC Set

The position and orientation characteristic (POC) set equations for the serial mechanism and parallel mechanism are expressed as respectively:

$$M_{bi} = \bigcup_{i=1}^{m} M_{Ji} \tag{1}$$

$$M_{Pa} = \overset{n}{\underset{i=1}{\cap}} M_{bi} \tag{2}$$

where

$M_{Ji}$—POC set generated by the *i*th joint.

$M_{bi}$—POC set generated by the end link of the *i*th chain.

$M_{Pa}$—POC set generated by the moving platform of PM.

### 3.2. Determining the DOF

The proposed general and full-cycle degrees of freedom (DOF) formula for PMs is given below:

$$F = \sum_{i=1}^{m} f_i - \sum_{i=1}^{v} \xi_{Lj} \tag{3}$$

$$\xi_{L_j} = \text{dim.}\left\{ \left( \overset{j}{\underset{i=1}{\cap}} M_{b_i} \cup M_{b_{(j+1)}} \right) \right\} \tag{4}$$

where

$F$—DOF of PM.

$f_i$—DOF of the *i*th joint.

$v$—number of independent loops, and $v = m - n + 1$.

$m$, $n$—number of all joints and links of the whole PM, respectively.

$\xi_{L_j}$—number of independent displacement equations (NIDE) of the *j*th loop.

$\overset{j}{\underset{i=1}{\cap}} M_{b_i}$—POC set generated by the sub-PM formed by the former *j* branches.

$M_{b(j+1)}$—POC set generated by the end link of (*j* + 1)th sub-chains.

### 3.3. Determining the Coupling Degree

According to the mechanism composition principle based on single-opened-chains (SOC) units, any PM can be decomposed into groups of SKC, and an SKC with $v$ independent loops can be decomposed into $v$ SOC. The constraint degree, denoted as $\triangle_j$, of the *j*th SOC is defined by:

$$\Delta_j = \sum_{i=1}^{m_j} f_i - I_j - \xi_{L_j} = \begin{cases} \Delta_j^- = -5, -4, -2, -1 \\ \Delta_j^0 = 0 \\ \Delta_j^+ = +1, +2, +3, \cdots \end{cases} \tag{5}$$

where

$m_j$—number of joints contained in the *j*th SOC$_j$.

$I_j$—the number of actuated joints in the *j*th SOC$_j$.

For an SKC, the following equation must be satisfied:

$$\sum_{j=1}^{v} \Delta_j = 0 \tag{6}$$

Then, the coupling degree of an SKC is expressed as:

$$k = \left| \Delta_j^- \right| = \Delta_j^+ = \frac{1}{2}\min\{\sum_{j=1}^{v} |\Delta_j|\} \tag{7}$$

Here, min{•} operation means that the decomposition sequence with the smallest $\sum\limits_{j=1}^{v} |\Delta_j|$ should be selected.

The constraint degree of the *jth* SOC indicates the constraint influences of the chain on the kinematic performance of the mechanism. Its physical meaning will be explained below.

(a) An SOC with a negative constraint degree, denoted as SOC$^-$, will apply $\left|\Delta_j^-\right|$ constraint equations to a mechanism, and the number of DOF of the mechanism will be decreased by DOF$_s$ of $\left|\Delta_j^-\right|$.

(b) An SOC with a positive constraint degree, denoted as SOC$^+$, will increase the number of DOF of the mechanism by $\Delta_j^+$. Therefore, its forward kinematics solutions could not be solved immediately. Its assembly can be determined only on the condition that $\Delta_j^+$ virtual variables are assigned. When the number of the virtual variables is equal to the number of $\left|\Delta_j^-\right|$ constraint equations applied in SOC$^-$, i.e., $k = \left|\Delta_j^-\right| = \Delta_j^+$, the motion of the mechanism is defined, and its forward kinematics can be obtained.

(c) An SOC with zero constraint degree, denoted as SOC$^0$, does not affect the *DOF*. Its forward kinematics solutions can be obtained immediately without assigning virtual variables.

Therefore, the coupling degree $k$ describes the complexity level of the topological structure of a PM, and it also represents the complexity level of its kinematic and dynamic analysis. The lower the coupling degree $k$, the easier the treatment of its forward kinematic and dynamic analysis [9,10].

The detailed explanations for these concepts and notations of topological characteristic index used for topological property analysis can be found in Ref. [9,12].

## 4. Case Studies

### 4.1. Three-Translation PM (3T-CU)

The three-translation PM proposed by authors [13], denoted as 3T-CU, as shown in Figure 1a, consists of a base platform 0, a moving platform 1, and three different branch chains. Among them, the first branch is $R_{11}//R_{21}//C_{31}$, which is connected to the base platform 0 through the revolute joint $R_{11}$ and connected to the moving platform 1 through the cylindrical joint $C_{31}$. The second branch is $R_{12}$-$U_{22}$-$U_{32}$, which is connected to the base platform 0 through the revolute joint $R_{12}$ and connected to the moving platform 1 through the moving joint $U_{32}$. The third branch is a hybrid branch chain, which includes a parallelogram, and its two ends are respectively connected to the base platform 0 and the moving platform 1 through revolute joints $R_{13}$ and $R_{33}$. Here, $R_{11}$, $R_{12}$, and $R_{13}$ on the base platform 0 could be the actuated joints.

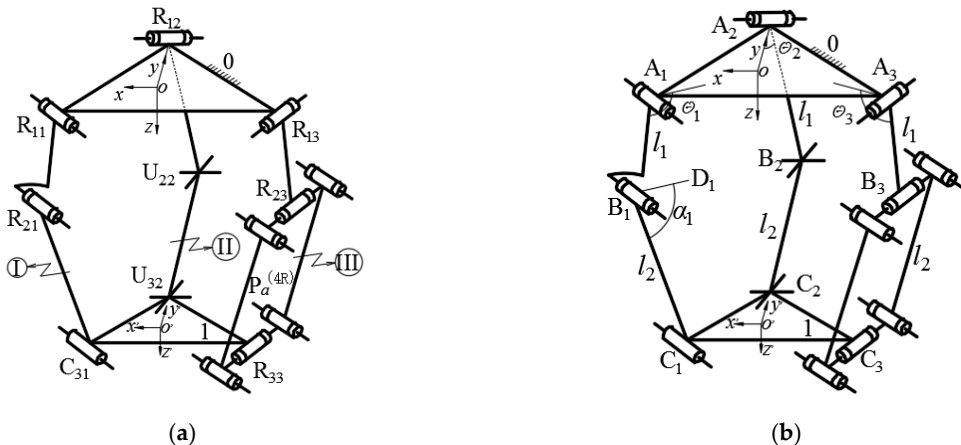

(**a**)                                  (**b**)

**Figure 1.** Three-translation parallel mechanisms (PM) (3T-CU). (**a**) Topology structure; (**b**) Kinematic modeling of the 3T PM.

4.1.1. Topology Analysis

(i) Analysis of the POC Set

Obviously, the topological architecture of chain I, II, and III of the PM can be denoted as, respectively:

$$SOC_1:\{R_{11}//R_{21}//C_{31}\},\ SOC_2:\{R_{12}\text{-}U_{22}\text{-}U_{32}\},\ SOC_3:\{R_{13}//R_{23}(P_a^{(4R)})//R_{33}\}.$$

Here, symbol "//" stands for "parallel", while symbol "-" stands for no special geometrical relation, which applies to the whole context of the paper.

The POC sets of the end of the three $SOC_S$ are determined according to Equation (1) as follows, respectively.

$$M_{SOC_1} = \begin{bmatrix} t^3 \\ r^1(\|R_{11}) \end{bmatrix},\ M_{SOC_2} = \begin{bmatrix} t^3 \\ r^2 \end{bmatrix},\ M_{SOC_3} = \begin{bmatrix} t^3 \\ r^1(\|R_{13}) \end{bmatrix}.$$

The POC set of the moving platform of this PM is determined from Equation (2) by

$$M_{P_a} = M_{SOC_1} \cap M_{SOC_2} \cap M_{SOC_3} = \begin{bmatrix} t^3 \\ r^1(\|R_{11}) \end{bmatrix} \cap \begin{bmatrix} t^3 \\ r^2 \end{bmatrix} \cap \begin{bmatrix} t^3 \\ r^1(\|R_{13}) \end{bmatrix} = \begin{bmatrix} t^3 \\ r^0 \end{bmatrix}.$$

Hence, the moving platform 1 of the PM has a three-translation motion output.

(ii) Determining the DOF

It is easy for this PM to have two selection methods for PAR, as follows,

Case ①: If the first loop of the PM (that is, sub-PM) consists of the first and third branch chains, namely $R_{11}//R_{21}//C_{31}\text{-}R_{33}//(P_a^{(4R)})R_{23}//R_{13}$, the number of independent displacement equations (NIDE) is obtained by Equation (4)

$$\xi_{L_1} = \dim\left\{ \begin{bmatrix} t^3 \\ r^1(\|R_{11}) \end{bmatrix} \cup \begin{bmatrix} t^3 \\ r^1(\|R_{13}) \end{bmatrix} \right\} = \dim\left\{ \begin{bmatrix} t^3 \\ r^2\|(\diamond(R_{11},R_{13})) \end{bmatrix} \right\} = 5.$$

From Equation (3), the degree of freedom (DOF) of the first loop (1st sub-PM) is:

$$F_{(1-2)} = \sum f - \xi_{L_1} = (4+4) - 5 = 3.$$

The second loop is composed of the above-mentioned sub-PM and the second branch chain. From Equation (4), the NIDE is calculated by

$$\xi_{L_2} = \dim\left\{ \begin{bmatrix} t^3 \\ r^1(\|R_{11}) \end{bmatrix} \cap \begin{bmatrix} t^3 \\ r^1(\|R_{13}) \end{bmatrix} \cup \begin{bmatrix} t^3 \\ r^2 \end{bmatrix} \right\} = \dim\left\{ \begin{bmatrix} t^3 \\ r^2 \end{bmatrix} \right\} = 5.$$

Case ②: If the first loop (sub-PM) of the PM consists of the second and the third branch chains, namely $R_{12}\text{-}U_{22}\text{-}U_{32}\text{-}R_{33}//(P_a^{(4R)})R_{23}//R_{13}$, the NIDE is obtained by Equation (4):

$$\xi_{L_1} = \dim\left\{ \begin{bmatrix} t^3 \\ r^2 \end{bmatrix} \cup \begin{bmatrix} t^3 \\ r^1(\|R_{13}) \end{bmatrix} \right\} = \dim\left\{ \begin{bmatrix} t^3 \\ r^2 \end{bmatrix} \right\} = 5.$$

From Equation (3), the degree of freedom of the first loop (1st sub-PM) is:

$$F_{(1-2)} = \sum f - \xi_{L_1} = (5+4) - 5 = 4.$$

The second loop is composed of the above-mentioned 1$^{\text{st}}$ sub-PM and the first branch chain. From Equation (4), the number of independent displacement equation is:

$$\xi_{\text{L}_2} = \dim\left\{\left[\begin{array}{c} t^3 \\ r^2 \end{array}\right] \cap \left[\begin{array}{c} t^3 \\ r^1(\|\text{R}_{13}) \end{array}\right] \cup \left[\begin{array}{c} t^3 \\ r^1(\|\text{R}_{11}) \end{array}\right]\right\} = \dim\left\{\left[\begin{array}{c} t^3 \\ r^2(\|\diamond(\text{R}_{11}, \text{R}_{13}) \end{array}\right]\right\} = 5.$$

Thus, the DOF of the PM is calculated from Equation (3) as:

$$F = \sum_{i=1}^{m} f_i - \sum_{i=1}^{v} \xi_{\text{L}_j} = (8+5) - (5+5) = 3.$$

Therefore, when the revolute joints $\text{R}_{11}$, $\text{R}_{12}$, and $\text{R}_{13}$ on the base platform 0 are selected as the actuated joints, the moving platform 1 can realize 3T motion outputs.

(iii) Determining the coupling degree

Form Equation (5), the constraint degree of the two loops are respectively given by:

$$\text{For Case : } \Delta_1 = \sum_{i=1}^{m_1} f_i - I_1 - \xi_{\text{L}_1} = 8 - 2 - 5 = +1, \Delta_2 = \sum_{i=1}^{m_2} f_i - I_2 - \xi_{\text{L}_2} = 5 - 1 - 5 = -1.$$

From Equation (6), the PM contains one SKC. Further, from Equation (7), the coupling degree of the SKC is given by

$$\kappa = \frac{1}{2}\min\{\sum_{j=1}^{v}|\Delta_j|\} = \frac{1}{2}(|+1|+|-1|) = 1.$$

Therefore, the coupling degree of the PM is $\kappa = 1$, which means that one virtual variable needs to be assigned when solving its positions.

$$\text{For Case ② } \Delta_1 = \sum_{i=1}^{m_1} f_i - I_1 - \xi_{\text{L}_1} = 9 - 2 - 5 = +2, \Delta_2 = \sum_{i=1}^{m_2} f_i - I_2 - \xi_{\text{L}_2} = 4 - 1 - 5 = -2.$$

From Equation (6), the PM contains one SKC. Furthermore, from Equation (7), the coupling degree of the SKC is given by

$$\kappa = \frac{1}{2}\min\{\sum_{j=1}^{v}|\Delta_j|\} = \frac{1}{2}(|+2|+|-2|) = 2$$

This moment, the coupling degree of the PM is $\kappa = 2$, which means that two virtual variables need to be assigned when solving its positions, and it undoubtedly makes the FPS much more complicated than the case ① with only one virtual variable.

(iv) Optimization selection for PAR

So far, it can be found that according to the cases ① and ②, the NIDE of the two cases obtained are the same, i.e., $\xi_{L1} = \xi_{L2} = 5$, but their constraint degree $\Delta$ (or coupling degree $\kappa$) is different, i.e., $\kappa$ of the case ① is one, while $\kappa$ of the case ② is two. Therefore, according to the optimization criteria for the PAR, the case ① that has the smallest constraint degree value ($\Delta_{\text{min}} = 1$) and the minimum NIDE ($\xi_{\text{min}} = 5$) should be used to solve the FPS. The details are described below.

4.1.2. Position Analysis

(i) The coordinate system and parameterization

The kinematic modelling of the PM is shown in Figure 1b. The base platform 0 is an equilateral triangle with a circle radius $R$, and select the geometric center O as the origin of the base coordinate system. The $x$- and $y$-axis are perpendicular and parallel to the line $OA_2$. Let moving platform 1 be an equilateral triangle with a circle radius $r$, and select the O′ point on the moving platform as the origin

of the moving coordinate system. The $x'$- and $y'$-axis are perpendicular and parallel to the line $O'C_2$. The $z$ and $z'$ axis are determined by the Cartesian coordinate rule.

Let the angle $\theta_i$ between vectors $A_iB_i$ and $A_iO$ be the input angle, and the length of the line $A_iB_i$ and $B_iC_i$ $(i = 1–3)$ is equal to $l_1$ and $l_2$, respectively.

(ii) Direct kinematics

To perform the FPS, i.e., it is to compute the position $O'(x, y, z)$ of the moving platform with the known actuated joints $\theta_1$, $\theta_2$, and $\theta_3$.

The coordinates of points $A_i$ and $B_i$ $(i = 1–3)$ are easily known as:

$$A_1 = (R\cos 30°, -R\sin 30°, 0)^{\mathrm{T}}, A_2 = (0, R, 0)^{\mathrm{T}}, A_3 = (-R\cos 30°, -R\sin 30°, 0)^{\mathrm{T}},$$
$$B_1 = ((R - l_1\cos \theta_1)\cos 30°, -(R - l_1\cos \theta_1)\sin 30°, l_1\sin \theta_1)^{\mathrm{T}},$$
$$B_2 = (0, R - l_1\cos \theta_2, l_1\sin \theta_2)^{\mathrm{T}}, B_3 = (-(R - l_1\cos \theta_3)\cos 30°, -(R - l_1\cos \theta_3)\sin 30°, l_1\sin \theta_3)^{\mathrm{T}}.$$

① Solving the first loop $(A_1\text{-}B_1\text{-}C_1\text{-}C_3\text{-}B_3\text{-}A_3)$ with a positive constraint $(\Delta_1 = 1)$

Since the coupling degree of the PM is 1, it is necessary to assign one virtual variable when performing the FPS. Let the angle $\alpha_1$ between $B_1C_1$ and $B_1D_1$ be the virtual variable, where $B_1D_1//A_1O$. It is easy to know that the coordinates of point $C_1$ are below:

$$C_1 = (x_{B_1} - l_2\cos 30°\cos \alpha_1 , y_{B_1} + l_2\sin 30°\cos \alpha_1 , z_{B_1} + l_2\sin \alpha_1)^{\mathrm{T}}. \tag{8}$$

Thus, the three points $C_i(i = 1–3)$ in the base coordinate system are calculated as:

$$C_1 = (r\cos 30° + x, -r\sin 30° + y, z)^{\mathrm{T}} \tag{9}$$

$$C_2 = (x, r + y, z)^{\mathrm{T}} \tag{10}$$

$$C_3 = (-r\cos 30° + x, -r\sin 30° + y, z)^{\mathrm{T}}. \tag{11}$$

From Equations (9) and (11), we can get:

$$C_3 = (x_{B_1} - l_2\cos 30°\cos \alpha_1 - 2r\cos 30° , y_{B_1} + l_2\sin 30°\cos \alpha_1 , z_{B_1} + l_2\sin \alpha_1)^{\mathrm{T}}. \tag{12}$$

Due to the length constraint defined by $B_3C_3 = l_2$, we can get:

$$\alpha_1 = 2\arctan\left(\frac{-G_2 \pm \sqrt{G_2{}^2 + G_1{}^2 - G_3{}^2}}{G_3 - G_1}\right) \tag{13}$$

where

$$p = x_{B_1} - 2r\cos 30° - x_{B_3}, q = y_{B_1} - y_{B_3}, g = z_{B_1} - z_{B_3}, G_1 = 2l_2(q\sin 30° - p\cos 30°),$$
$$G_2 = 2gl_2, G_3 = p^2 + q^2 + g^2.$$

Substituting Equation (13) into Equations (8) and (12), the points $C_1$ and $C_3$ can be obtained.

② Solving the second loop $(Loop_2: A_2\text{-}B_2\text{-}C_2)$ with negative constraint $(\Delta_2 = -1)$

After the positions of all joints on the first loop are obtained, it is easier to solve that on the second loop. From Equations (9) and (10), the coordinates of point $C_2$ are:

$$C_2 = (x_{C_1} - r\cos 30° , y_{C_1} + r + r\sin 30° , z_{C_1})^{\mathrm{T}}. \tag{14}$$

Therefore, the origin coordinates of the moving platform can be easily obtained.

It can be seen that from FPS of the PM there is a total of five position equations, i.e., (1) three geometric constraint equations: $B_1C_1 = l_2$, $B_3C_3 = l_2$, $y_{C_3} = y_{C_1}$; (2) two topological constraint equations introduced by the POC feature (3T): $z_{C_2} = z_{C_1}$, $z_{C_3} = z_{C_1}$, which are exactly equal to the NIDE $\xi_L = 5$, and we ensure that the positions of the PM can be solved.

The inverse kinematics of this PM is omitted here due to its easiness.

(iii) Verification

Set the parameters of the PM as $R = 90$, $r = 55$, $l_1 = 40$, $l_2 = 40$(unit: mm), and the input angles of the three actuated joints are $\theta_1 = 30°$, $\theta_2 = 60°$, and $\theta_3 = 60°$.

From Equations (8)–(14), two sets of real solutions are obtained, i.e., (1) $x = -33.9339$, $y = 19.5917$, $z = 13.9672$, and (2) $x = 23.5901$, $y = -13.6197$, and $z = 49.6216$. It has been verified that both the FPS and inverse solutions derived above are correct.

The authors also analyze the position of the 3T-CU PM according to case ②, but only numerical solutions were obtained, and calculations are more complicated.

So far, for route selection cases ① and ②, the NIDE are $\xi_{L_1} = \xi_{L_2} = 5$, but the constraint degree of case ① is $\Delta_1 = +1, \Delta_2 = -1$ and the constraint degree of case ② is $\Delta_1 = +2, \Delta_2 = -2$. Therefore, case ① is the optimization selection for PAR. It not only guarantees the efficient kinematic modeling but also obtains the symbolic solutions.

### 4.2. Three-Translation PM (Delta-CU)

Figure 2a shows another three-translation PM [14,15], denoted as Delta-CU PM, designed by the authors, which consists of base platform 0, moving platform 1, and three branch chains. Among them, the first and third ones are hybrid branches containing a parallelogram structure (same as the traditional Delta mechanism). The first and third branches are connected to the base platform 0 through the revolute joints $R_{11}$ and $R_{13}$, and they are connected to the moving platform 1 through the revolute joints $R_{31}$ and $R_{33}$, respectively. The second branch is $R_{12}$-$U_{22}$-$U_{32}$, and its two ends are connected to base platform 0 and moving platform 1 through $R_{12}$ and $U_{32}$, respectively. The PM is called Delta-CU, i.e., 2-R//R//$P_a$//R+R-U-U.

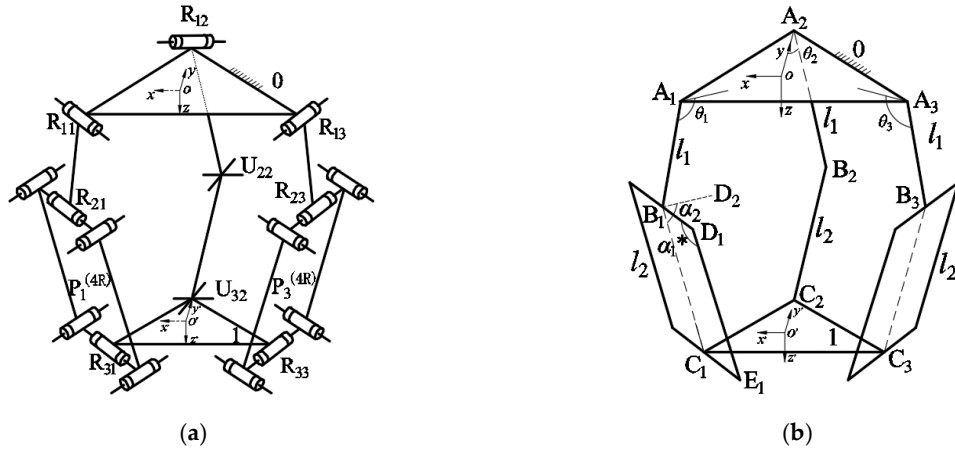

(a)          (b)

**Figure 2.** Three-translation PM (Delta-CU). (**a**) Topology structure; (**b**) Kinematic modeling of the 3T PM.

Compared with the Delta mechanism with the same three branches (all of which are hybrid branches with a parallelogram structure), this Delta-CU mechanism decreases two revolute joints and three links, and hence the structure is simpler. Since the topological structure of the three branch chains are not all different, it involves the problem of optimization selection of PAR, and different PARs will lead to a big difference in the difficulty of the FPS of the PM.

### 4.2.1. Topological Analysis

(i) Analysis of the POC Set

The first and third mixed branches are denoted as $SOC_i \{R_{1i}//R_{2i}//Pa^{(4R)}-R_{3i}\}(i = 1, 3)$, while the second branch is denoted as $SOC_2\{R_{12}-U_{22}-U_{42}\}$.

The POC sets of the end of the three $SOC_S$ are determined according to Equation (1) as follows, respectively:

$$M_{SOC_1} = \begin{bmatrix} t^3 \\ r^1(\|R_{11}) \end{bmatrix}, M_{SOC_2} = \begin{bmatrix} t^3 \\ r^2 \end{bmatrix}, M_{SOC_3} = \begin{bmatrix} t^3 \\ r^1(\|R_{13}) \end{bmatrix}$$

The POC set of the moving platform of this PM is determined from Equation (2) by:

$$M_{P_a} = M_{SOC_1} \cap M_{SOC_2} \cap M_{SOC_3} = \begin{bmatrix} t^3 \\ r^1(\|R_{11}) \end{bmatrix} \cap \begin{bmatrix} t^3 \\ r^2 \end{bmatrix} \cap \begin{bmatrix} t^3 \\ r^1(\|R_{13}) \end{bmatrix} = \begin{bmatrix} t^3 \\ r^0 \end{bmatrix}$$

Hence, the moving platform 1 of the PM has three-translation motion output.

(ii) Determining the DOF

The PM has three branches; therefore, it contains two independent loops. Since the three branches belong to two different topological structures, that is, the first and third branches are hybrid branches with four degrees of freedom (DOF) (one 4R parallelogram mechanism is equivalent to one prismatic joint). The DOF of the second branch is 5. Therefore, there are at least two schemes in the topology decomposition of the PM as follows.

Case ①: The first loop (sub-PM) is composed of the first and third hybrid branch chains with the least degree of freedom (DOF = 4), while the second loop is composed of the sub-PM and the second branch ($R_{12}$-$U_{22}$-$U_{32}$) with 5-DOF.

Case ②: The first loop (sub-PM) is composed of the first branch chain with 4-DOF and the second branch chain with 5-DOF, while the second loop is composed of the sub-PM and the third branch chain with 4-DOF.

For Case ①: The NIDE of the two loops can be obtained by Equation (4):

$$\xi_{L_1} = \dim \left\{ \begin{bmatrix} t^3 \\ r^1(\|R_{11}) \end{bmatrix} \cup \begin{bmatrix} t^3 \\ r^1(\|R_{13}) \end{bmatrix} \right\} = \dim \left\{ \begin{bmatrix} t^3 \\ r^2 \end{bmatrix} \right\} = 5, \xi_{L_2} = \dim \left\{ \begin{bmatrix} t^3 \\ r^1(\|R_{11}) \end{bmatrix} \cap \begin{bmatrix} t^3 \\ r^1(\|R_{13}) \end{bmatrix} \cup \begin{bmatrix} t^3 \\ r^2 \end{bmatrix} \right\} = \dim \left\{ \begin{bmatrix} t^3 \\ r^2 \end{bmatrix} \right\} = 5.$$

From Equation (3), the DOF of the PM is:

$$F = \sum_{i=1}^{m} f_i - \sum_{i=1}^{v} \xi_{L_j} = (8 + 5) - (5 + 5) = 3.$$

Therefore, when the revolute joints $R_{11}$, $R_{12}$, and $R_{13}$ on the base platform 0 are selected as the actuated joints, the moving platform 1 can realize 3T motion outputs.

For Case ②: The NIDE of the two loops can be obtained by Equation (4):

$$\xi_{L_1} = \dim \left\{ \begin{bmatrix} t^3 \\ r^1 \end{bmatrix} \cup \begin{bmatrix} t^3 \\ r^2 \end{bmatrix} \right\} = \dim \left\{ \begin{bmatrix} t^3 \\ r^3 \end{bmatrix} \right\} = 6, \xi_{L_2} = \dim \left\{ \begin{bmatrix} t^3 \\ r^1 \end{bmatrix} \cap \begin{bmatrix} t^3 \\ r^2 \end{bmatrix} \cup \begin{bmatrix} t^3 \\ r^1 \end{bmatrix} \right\} = \dim \left\{ \begin{bmatrix} t^3 \\ r^1 \end{bmatrix} \right\} = 4.$$

From Equation (3), the DOF of the PM is:

$$F = \sum_{i=1}^{m} f_i - \sum_{i=1}^{v} \xi_{L_j} = (4 + 5 + 4) - (5 + 5) = 3.$$

The DOF of the PM is still three. Therefore, when $R_{11}$, $R_{12}$, and $R_{13}$ on the base platform are actuated joints, the moving platform 1 can realize 3T motion outputs.

(iii) Determining the coupling degree

Case ①: From Equation (5), the constraint degrees of the two loops are respectively given by:

$$\Delta_1 = \sum_{i=1}^{m_1} f_i - I_1 - \xi_{L_1} = (4 + 4) - 2 - 5 = +1, \Delta_2 = \sum_{i=1}^{m_2} f_i - I_2 - \xi_{L_2} = 5 - 1 - 5 = -1.$$

From Equation (6), the PM contains one SKC. Furthermore, from Equation (7), the coupling degree of the SKC is given by:

$$k = \frac{1}{2}\min\{\sum_{j=1}^{v}|\Delta_j|\} = \frac{1}{2}(|+1|+|-1|) = 1.$$

The above equation shows the following: ① This PM contains only one SKC, and its coupling degree is one. When performing FPS, it is necessary to assign one virtual variable on the first loop with the constraint degree of positive one, and it is necessary to establish a constraint position equation containing the virtual variable on the second loop with a constraint degree of minus one. ② Since the three actuated joints are in one SKC, according to the input–output (I-O) motion decoupling determination principle [9], it can be determined that the PM does not have an input–output (I-O) motion decoupling property.

For Case ②: From Equation (5), the constraint degrees of the two loops are respectively given by:

$$\Delta_1 = \sum_{i=1}^{m_1}f_i - I_1 - \xi_{L_1} = (4+5) - 2 - 6 = +1, \Delta_2 = \sum_{i=1}^{m_2}f_i - I_2 - \xi_{L_2} = 4 - 1 - 4 = -1.$$

From Equation (6), the PM contains one SKC. Furthermore, from Equation (7), the coupling degree of the SKC is calculated as:

$$k = \frac{1}{2}\min\{\sum_{j=1}^{v}|\Delta_j|\} = \frac{1}{2}(|+1|+|-1|) = 1.$$

Of course, the topology decomposition scheme ② of the first loop of the PM can also be composed of the second and third branches, while the second loop is composed of the sub-PM and the first branch, and the result is identical.

(iv) Optimization selection for PAR

Comparing the topological decomposition Cases ① and ②, it can be seen that if the position analysis is performed according to Case ②, the constraint degree of the two loops are $\Delta_1 = 1$ and $\Delta_2 = -1$, and the NIDEs are $\xi_{L1} = 6$, $\xi_{L2} = 4$, respectively, which means that when performing the FPS, six position constraint equations must be found in the first loop. Obviously, the difficulty will increase, and it will even not be solved. On the contrary, if the position analysis is carried out according to Case ①, the constraint degrees of the two loops are $\Delta_1 = 1$, $\Delta_2 = -1$, and the NIDEs are $\xi_{L1} = 5$ and $\xi_{L2} = 5$, respectively, which means that when performing the FPS, only five position constraint equations are needed to be found in the first loop, which is obviously easier.

Therefore, when performing the FPS of the Delta-CU PM, the PAR should be selected as Case ①.

4.2.2. Position Analysis

(i) The coordinate system and parameterization

The kinematic modeling of the Delta-CU PM is shown in Figure 2b. The base platform 0 is an equilateral triangle with a circle radius $R$, and select the geometric center O as the origin of the base coordinate system. The $x$- and $y$-axis are perpendicular and parallel to the line $OA_2$. Let moving platform 1 be an equilateral triangle with a circle radius $r$, and select the O′ point on the moving platform as the origin of the moving coordinate system. The $x'$- and $y'$-axis are perpendicular and parallel to the line $O'C_2$. The $z$- and $z'$-axis are all determined by the Cartesian coordinate rule. Let the angle $\theta_i$ between vectors $A_iB_i$ and $A_iO$ be the input angle, and the lengths of the lines $A_iB_i$ and $B_iC_i$ ($i = 1$–3) are equal to $l_1$ and $l_2$, respectively. Let the coordinates of the origin of the moving coordinate system be O′ $(x, y, z)$.

(ii) Direct kinematics

To perform the FPS, compute the position O′ $= (x, y, z^*)$ of the platform with the known actuated joints $\theta_2$, $\theta_2$, and $\theta_3$.

Since the coupling degree of the PM is $\kappa = 1$, one virtual variable needs to be assigned. Furthermore, the first loop passes through the moving platform 1, and there are two methods for selecting virtual variables, namely method A (i.e., the virtual variable starts from one side of the loop) and method B (i.e., the virtual variable starts from the moving platform) [16]. Since method B has the advantages of a short calculation path, fewer calculations, and available symbolic solutions [16], method B is now used for the calculation of the PM.

Since the PM is a 3T PM and the coupling degree is $\kappa = 1$, take one of the position parameters $(x, y, z^*)$ of the platform 1, for example, $z^*$, as a virtual variable.

① Solving the first loop with the positive constraint ($\Delta_1 = 1$)

The coordinates of $C_i$ point ($i = 1$–3) on the moving platform are given, respectively.

$$C_1 = (r\cos 30° + x, -r\sin 30° + y, z*)^{\mathrm{T}}, C_2 = (x, r + y, z*)^{\mathrm{T}}, C_3 = (-r\cos 30° + x, -r\sin 30° + y, z*)^{\mathrm{T}}.$$

Due to the length constraint defined by $B_iC_i = l_2$ ($i = 1, 3$), we can get:

$$(x + m)^2 + (y + n)^2 + (z* - z_{B_1})^2 = l_2{}^2 \tag{15}$$

$$(x + u)^2 + (y + v)^2 + (z* - z_{B_3})^2 = l_2{}^2 \tag{16}$$

where

$$m = r\cos 30° - x_{B_1}, n = -\left(r\sin 30° + y_{B_1}\right), u = -\left(r\cos 30° + x_{B_3}\right), v = -\left(r\sin 30° + y_{B_3}\right).$$

② Solve the second loop with the negative constraint ($\Delta_2 = -1$)

Due to the length constraint defined by $B_2C_2 = l_2$, we can get:

$$(x - x_{B_2})^2 + (y + t)^2 + (z* - z_{B_2})^2 = l_2{}^2 \tag{17}$$

where $t = r - y_{B_2}$.

From Equations (15)–(17), we can get the symbolic solutions as follows:

$$\begin{cases} x = Nz* + D \\ y = Hz* + K \\ z* = \dfrac{-C_2 \pm \sqrt{C_2{}^2 - 4C_1C_3}}{2C_1} \end{cases} \tag{18}$$

where

$$M = \frac{m^2 + n^2 + z_{B_1}{}^2 - x_{B_2}{}^2 - t^2 - z_{B_2}{}^2}{2}, T = m^2 + n^2 + z_{B_1}{}^2 - u^2 - v^2 - z_{B_3}{}^2 - \frac{2M(m-u)}{m + x_{B_2}}, P = \frac{2(m-u)(t-n)}{m + x_{B_2}} + 2(n - v),$$

$$Q = \frac{2(m-u)\left(z_{B_1} - z_{B_2}\right)}{m + x_{B_2}} + 2\left(z_{B_3} - z_{B_1}\right), H = -\frac{Q}{P}, K = -\frac{T}{P},$$

$$N = -\frac{(n-t)H + \left(z_{B_2} - z_{B_1}\right)}{x_{B_2} + m}, D = -\frac{(n-t)K + M}{x_{B_2} + m},$$

$$C_1 = N^2 + H^2 + 1, C_2 = 2\left(ND + HK + mN + nH - z_{B_1}\right), C_3 = D^2 + K^2 + 2mD + 2nK + m^2 + n^2 + z_{B_1}{}^2 - l_2{}^2.$$

It can be seen that from the FPS of the PM there is also a total of five position equations in the first loop, i.e., ① two position constraint equations $z_{C_1} = z_{C_3} = z$ introduced by the topological constraint that the moving platform 1 always performs three-dimensional translation, ② two length constraint conditions $B_1C_1 = l_2$ and $B_3C_3 = l_2$, and ③ the position of $C_2$ is also constrained by the length constraint equation $B_2C_2 = l_2$ inside the second loop. In this way, there are five position constraint equations, which are equal to the NIDE $\zeta_L = 5$. Therefore, the position of the first loop is guaranteed to be solvable, since these position equations are nonlinear ones and simper, symbolic solutions can be easily obtained.

The inverse kinematics of this PM and numerical verification are omitted here due to its easiness.

## 5. Conclusions

The optimization selection for PAR can affect the effectiveness of FPS and the solution forms. For this reason, the optimization selection criteria for PAR are proposed, i.e., when solving the position, the loop should be selected according to the criteria of "minimum constraint degree value ($\Delta_{min}$) and minimum number of independent displacement equations ($\xi_{min}$)" in order to effectively perform FPS and obtain the symbolic solutions to the greatest possible extent. Otherwise, the FPS may be difficult, complicated, or symbolic solutions cannot be obtained. Two examples are used to illustrate the procedures of how to select the PAR.

In the recent years, the authors have performed a large number of FPS of the PMs [17,18], which proves that the procedures of the selection criteria of PAR for FPS proposed in this paper are universal.

For PMs with branch chains of different topology, the correct selection of the PAR is necessary and important. This work provides new inspiration and the road map for the FPS of these complex PMs.

**Author Contributions:** Conceptualization, H.S. and T.-l.Y.; methodology, Q.X. and J.L.; validation, Q.X.; writing—original draft preparation, H.S.; writing—review and editing, Q.X. and J.L.; funding acquisition, H.S. All authors have read and agreed to the published version of the manuscript.

**Funding:** This research is sponsored by the NSFC (No. 51975062, 51375062).

**Conflicts of Interest:** The authors declare no conflict of interest.

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
