# Peer review of "The Effect of the Optimization Selection of Position Analysis Route on the Forward Position Solutions of Parallel Mechanisms"

_robotics, doi:10.3390/robotics9040093_

Round 1

Reviewer 1 Report

The paper is very interesting as the authors proposes the coupling degree as an index that greatly facilitate the resolution of forward kinematic problem for parallel mechanism. The index allows identifying the first kinematic chain solve. The reviewer suggests that the following comments shall be addressed to improved the manuscript.

Major comments:

  • Several variables are not explained. Examples: "n" under Eq. (4), "||", "-" and the diamond symbol in 4.1.1.i and 4.2.1.i.
  • Is there any reference on the resolution of the forward kinematics for the mechanism illustrated?
  • It is claimed that identifying the kinematic chain with the lowest coupling degree simplify the resolution of the forward kinematics, but this cannot be verified as no comparison with other method is provided.
  • Following the previous comment, how does this compare in terms of complexity with a resolution using a another kinematic chain which does not have the lowest coupling degree?

Minor comments:

  • The logo and title of the journal "Sensors" is wrongly used at the first page.
  • All equations shall be number to facilitate the reading.
  • Page 1, line 41: "with different topology of branch chains" shall be "with branch chains of different topology".
  • The last sentence of paragraph 2.1 and first sentence of paragraph 2.2 are redundant.
  • The reviewer could not understand the sense of the paragraph 2.2.i. 
  • The article "a" should be written "an" in page 2, line 64
  • Spaces are missing before some bracket and parenthesis.

Reviewer 2 Report

The paper is relatively clear and well structured. It deals with the forward position solution of complex parallel mechanisms, and more precisely with the optimization selection criterion of position analysis route in this context. The problem is rather classical, but the proposed approach is rather pertinent. The paper is interesting and globally convenient, but could be improved with some slight modifications. The detail is presented in the attached file.

With these light modifications, the paper can be published.

Round 2

Reviewer 1 Report

Paper is now good enough for the journal standard.